# Leaf Extracts of Invasive Woody Species Demonstrate Allelopathic Effects on the Growth of a Lawn Grass Mixture

**DOI:** 10.3390/plants12244084

**Published:** 2023-12-06

**Authors:** Olga V. Shelepova, Ekaterina V. Tkacheva, Aleksandr A. Ivanovskii, Ludmila V. Ozerova, Yulia K. Vinogradova

**Affiliations:** 1Plant Physiology and Immunity Laboratory, N.V. Tsitsin Main Botanical Garden of Russian Academy of Sciences, Botanicheskaya 4, Moscow 127276, Russia; shov_gbsad@mail.ru; 2Faculty of Biology, Shenzhen MSU-BIT University, International University Park Road 1, Dayun New Town, Longgang District, Shenzhen 517182, China; ivanovskii_a@smbu.edu.cn; 3Faculty of Biology, Lomonosov Moscow State University, Moscow 119991, Russia; 4Plant Tropical Laboratory, N.V. Tsitsin Main Botanical Garden of Russian Academy of Sciences, Botanicheskaya 4, Moscow 127276, Russia; lyozerova@yandex.ru; 5Laboratory of Natural Flora, N.V. Tsitsin Main Botanical Garden of Russian Academy of Sciences, Botanicheskaya 4, Moscow 127276, Russia; gbsad@mail.ru

**Keywords:** invasive wood species, leaf litter, secondary metabolites, macro- and microelements, allelopathic activity

## Abstract

Biochemical composition was studied in the leaf litter of alien woody species included in the 100 most aggressive invasive species of Europe: *Ailanthus altissima*, *Quercus rubra*, *Acer negundo*, *Robinia pseudoacacia*, and *Elaeagnus angustifolia*. Using GC-MS, we detected 187 metabolites in the leaf litter, which are phenolic acids and their derivatives, carbohydrates and their derivatives, polyphenolic compounds, cyclic esters, glycosides, and amino acids and their derivatives. Species-specific metabolites were identified for each species. The main allelochemicals in the leaf litter extract of *Q. rubra* are determined mainly by the relative abundance of phenolic and fatty acids and their esters, whereas those in the leaf litter extract of *R. pseudoacacia* are determined by carbohydrates and their derivatives and ester of fatty acid, and those in the leaf litter extract of *A. altissima* are determined by glycosides. Profiles of macro- and microelements were characterized. It was found that aqueous extracts of leaf litter from all the invasive woody plants under study have a negative effect on the seed germination and initial growth of *Vicia cracca* and *Avena strigosa* used for the reclamation of disturbed urban and industrial lands. At the same time, *V. cracca* is potentially more sensitive.

## 1. Introduction

Invasive species utilize a wide array of trait strategies to establish in novel ecosystems. One of the hypotheses explaining the successful invasion of a secondary geographic range by alien plant species is the “novel weapons” hypothesis [1,2,3]. It suggests that some alien species possess biochemicals with potent allelopathic activity or may create new interactions between plants and soil microorganisms. These substances, owing to a long period of adaptation, are relatively ineffective against neighboring species in a natural geographic range and can significantly inhibit the growth of plants in the communities affected by the invasion of a new habitat. Moreover, the selective advantage of having these products may lead to the rapid evolution of this trait, for example, to the production of larger amounts of allelopathic exudates [4,5].

There is quite a lot of evidence supporting the hypothesis that the majority of invasive plant species produce allelochemicals with the potential to negatively affect native plant performance [3,6,7]. Non-native plant species are known to affect the availability and uptake of nutrients, alter the population growth rates of native species, and affect the abundance of individual species within a community [7,8]. Unfortunately, the lack of uniformity in the screening of potentially allelopathic chemicals means that the extent to which the same species exhibits allelopathy can vary among studies [3].

Research on woody leaf litter can yield more unambiguous results because in the autumn, under invasive trees, fallen leaves cover the undercrown space in three to four layers. For example, communities of woody invasive species (*Ailanthus altissima* L. and *Robinia pseudoacacia* L.) are reported to show additional peaks of litter precipitation in the late spring or summer, and annual litter volume is twice as high as that in their forests of origin [9,10]. Low-quality litter (a low nitrogen content and a high C/N ratio) is associated with changes in the physicochemical parameters of soil in the forests consisting of the invasive woody species *Quercus rubra* L. [11]. The soil under *Q. rubra* is characterized by significantly lower water-holding capacity and lower concentrations of organic carbon, total phosphorus, magnesium, calcium, ammonia and nitrate forms of nitrogen, total phenolic compounds, and condensed tannins as compared with uninvaded plant communities.

We believe that it is the allochthonous organic substances entering ecosystems through fallen leaves that are responsible for the new biochemical impact on neighboring species, especially grass-layer species. Leaf litter is one of the most important sources of the nutrients and energy that support food chains in oligotrophic systems [12,13] and can also serve as an important source of the nutrients that cause siltation processes along banks of water bodies and streams in the littoral zone of large bodies of water [14,15]. An influence of ground litter on the mineralization of nutrients was noticed in many other articles [16,17], indicating that species richness and/or the composition of ground litter are important for processes for decomposition and the functioning of ecosystems. That is why it is important and relevant to identify the specific compounds, such as polyphenols and macro- and microelements, that may be involved in the decomposition process either as inhibitors or as stimulants.

There is evidence that the decomposition rate of leaves of invasive species is higher than that of native species; this rate is 2.6 times higher in herbaceous plants and 1.3 times higher in woody plants [18]. Therefore, for a comparative study on leaf litter, it is advisable to choose species of the same life form. In our current work, we chose woody invasive species because trees are the main component that supports the functioning of forest plant communities [19].

The purpose of our study was to qualitatively profile the secondary metabolites and microelements in the leaf litter of the woody invasive species that are the most aggressive in Europe as well as to evaluate the influence of the leaf litter on the germination of seeds and growth of seedlings of two test plants.

## 2. Results

### 2.1. Metabolic Analysis

A total of 187 metabolites were identified in the leaf litter water extracts (Appendix A). The metabolites were categorized into the following classes: organic acids (**55** compounds), carbohydrates and their derivatives (**57** compounds), polyphenolic compounds (**34** compounds), cyclic esters (**4** compounds), glycosides (**4** compounds), and amino acids and their derivatives (**6** compounds). Below, these groups are referred to as “Phenolic and aliphatic acids”, “Carbohydrates”, “Polyphenols”, “Esters”, “Glycosides”, and “Amino acids”, respectively. Differences among the profiles of the leaf litter water extracts were determined, and common metabolites and compounds specific to individual species were identified. Figure 1 and Figure 2 represent the species metabolite profiles of the leaf litter extracts of alien woody species for groups of metabolites and for the most common and abundant individual metabolites, respectively. The species metabolite profiles of the leaf litter extracts differed significantly both at the metabolite group level and at the individual metabolite level (PerMANOVA *p* < 0.01 for each case). At the pairwise level, differences between each pair of species were also significant (*p* < 0.05 in each case).

The largest number of compounds was identified in the leaf litter extract of *R. pseudoacacia*, and the smallest in the leaf litter extract of *Q. rubra*. The number of metabolites in the analyzed leaf litter extracts decreased in the following order: *R. pseudoacacia* (**142** compounds) > *A. altissima* (**73** compounds) > *E. angustifolia* (**41** compounds) > *A. negundo* (**36** compounds) > *Q. rubra* (**31** compounds).

It was found that the chemical composition of the leaf litter extracts was characterized by a substantial concentration of carboxylic acids and their derivatives, which accounted for 62.9 ± 1.3% to 19.2 ± 0.4% (wt.% of the extract), and their proportion decreased in the following order: *Q. rubra* (62.9 ± 1.3%) > *A. negundo* (41.4 ± 1.0%) > *E. angustifolia* (35.8 ± 0.5%) > *R. pseudoacacia* (19.2 ± 0.6%) > *A. altissima* (19.2 ± 0.4%). The largest number of carboxylic acids was identified in the leaf litter extract of *R. pseudoacacia*: 44 of these compounds were identified. In the leaf litter extract of *A. altissima*, 25 of these compounds were detected, whereas in the remaining three species, there were 14–11 carboxylic acids. Two acids (butanedioic acid and glucuronic acid) proved to be present in the leaf litter extracts of all five tree species. Most of the other acids were present in the leaf litter extracts of four or three tree species. Only two phenolic acids were dominant in the leaf litter extract of *Q. rubra*: glucopyranuronic acid (21.3 ± 0.5%) and protocatechoic acid (15.5 ± 0.7%), whereas in the leaf litter extract of *E. angustifolia*, pentanoic acid was dominant (11.8 ± 0.3%). In the leaf litter extract of *R. pseudoacacia*, the accumulation of dimethyl ester of hexanedioic acid was noted at 20.7 ± 0.4%.

The relative abundance of carbohydrates and their derivatives in the leaf litter extracts varied from 40.1 ± 0.9% to 15.2 ± 0.7% and decreased in the order: *R. pseudoacacia* (40.1 ± 0.9%) > *A. altissima* (28.0 ± 1.1%) > *Q. rubra* (20.1 ± 0.7%) > *E. angustifolia* (16.1 ± 0.5%) > *A. negundo* (15.2 ± 0.7%).

In this context, in the leaf litter extract of *R. pseudoacacia*, 22 carbohydrates were identified, whereas in the leaf litter extracts of *E. angustifolia* and *Q. rubra*, only nine of these compounds were detectable. Among carbohydrates, methyl-a-d-glucofuranoside and d-psicofuranose were found in the leaf litter extracts of all five species, and two monosaccharides—l-(-)-sorbose and d-(+)-glucosamine—were only found in the leaf litter extracts of *R. pseudoacacia* and *Q. rubra*, respectively.

The relative abundance of polyphenolic compounds in the leaf litter extracts ranged from 30.7 ± 2.2% to 15.4 ± 0.9% and decreased in the following order: *A. altissima* (30.7 ± 2.2%) > *E. angustifolia* (27.5 ± 2.3%) > *A. negundo* (22.8 ± 1.5%) > *Q. rubra* (16.3 ± 1.1%) > *R. pseudoacacia* (15.4 ± 0.9%). In the leaf litter extract of *R. pseudoacacia* leaves, 22 polyphenolic compounds were registered, while in the leaf litter extracts of *A. negundo*, *E. angustifolia,* and *Q. rubra*, only 8–9 polyphenolic compounds were found. Among the polyphenolic compounds, four sugar alcohols and glycerol dominated, which were found in the leaf litter extracts of all five tree species. It is worth noting that there was a considerable accumulation of adonitol (18.1 ± 0.4%) in the leaf litter extract of *A. altissima*; this compound is a pentahydric alcohol present in the cell wall of Gram-positive bacteria.

In addition, the leaf litter extracts contain glycosides, and in the leaf litter extract of *A. altissima*, a phenolic-type glycoside called arbutin accumulated in a very substantial amount: 13.8 ± 0.7%, whereas in the leaf litter extract of *A. negundo*, the accumulation of α-l-rhamnopyranose was noted at 6.3 ± 0.2%.

Nitrogen compounds had the lowest relative abundance among the identified organic compounds in the leaf litter extracts, and in this regard, the tree species could be ranked as follows: *E. angustifolia* (12.9 ± 0.8%) > *A. negundo* (9.6 ± 0.5%) > *R. pseudoacacia* (2.6 ± 0.4%) > *A. altissima* (0.6 ± 0.1%) > *Q. rubra* (0.9 ± 0.1%). In the leaf litter extracts of four tree species, five to six of these compounds were detectable, with the highest relative abundance of individual compounds being ~5%. Only in the leaf litter extract of *Q. rubra*, two of these compounds were identified at trace levels.

### 2.2. Analysis of Elemental Composition

The ash in the leaf litter of the analyzed trees was rich in macro- and microelements including O, C, K, Ca, Mg, P, Si, Mn, Mo, S, Al, and Zn. The levels of macro- and microelements varied substantially depending on the plant species (Appendix A). The species profiles by chemical element contents differed significantly (overall PerMANOVA *p* < 0.01, pairwise PerMANOVA *p* < 0.05 for each pair of species) (Figure 3).

In the leaf litter ash, oxygen dominated the total elemental composition; the highest relative abundance of this element was registered in the ash of *Q. rubra* leaf litter (43.7 ± 2.2%), and the lowest in the ash of *E. angustifolia* leaf litter (38.6 ± 1.8%). The C relative abundance was the highest in the ash of *Q. rubra* leaf litter (32.8 ± 1.4%), while in the ash of *R. pseudoacacia* leaf litter, this parameter was 1.8 times. The relative abundance level of Ca was 18.5–10.4%, K was −14.3–4.2%, and Mg was 8.3–2.7%, respectively.

In our work, the concentrations of P, Mn, Mo, S, and Si in the leaf litter ash did not exceed 2%, whereas Al and Zn constituted less than 1%. The content of each chemical element except O_2_ differed significantly among invasive species (*p* < 0.01 for each element either in the ANOVA or the Kruskal–Wallis test, see Section 4 for details). Among them, the undoubted leader in the accumulation of individual elements was *R. pseudoacacia*: the highest levels of five elements (O, K, Mg, Mn, and Mo) were registered in the leaf litter ash of this species, and the total relative abundance of these elements was 69.0 ± 2.3%. In the ash of *Q. rubra* leaf litter, the total relative abundance of these five elements was only 45.9 ± 1.4%, and only two elements (S and Zn) were dominant.

### 2.3. Assessment of the Allelopathic Activity of the Leaf Litter of the Invasive Woody Species toward Two Herbaceous Plants

#### 2.3.1. Assessment of the Allelopathic Activity of the Leaf Fall of Invasive Woody Species on the Seed Germination of Test Plants

This experiment indicated that the leaf litter extracts of alien woody species have a suppressive influence on the germination and growth of *V. cracca* and *A. strigosa*. These species are widely used for the reclamation of disturbed urban and industrial lands. They belong to the main phylogenetic lineages of vascular plants (monocotyledons and dicotyledons) and have different adaptations for soil nutrition, which was one of the criteria for their selection. In addition, related species of the test plants (*Vicia sativa* L. and *Avena sativa* L.) are often used in experiments to test the toxicity of certain compounds, which makes it possible to compare our results with those of other researchers. The autumn leaf litter of alien woody species, the composition of which was previously determined by us, was used in the experiment. This allowed us to estimate which groups of organic compounds that are included in the composition of autumn leaves of woody species have the maximum negative effect on the germination and further growth of test plants.

The extracts had the strongest toxic effect on the seeds of vetch (*V. cracca*). After soaking in metabolite solutions having a concentration 10 times less than that in the initial solution (mother liquor), the germination capacity (%) of vetch seeds was 5.0–9.5% lower in all experimental groups as compared with the control. After 6 days of the experiment, the toxic effect of the experimental extracts became even more noticeable: only four plants survived depending on the experimental group. The proportion of live plants was 98.5 ± 0.4–19.1 ± 0.6% as compared with the control and decreased in the following order: *A. negundo* (98.5 ± 0.4%) > *A. altissima* (95.2 ± 0.6%) > *R. pseudoacacia* (95.2 ± 0.4%) > *E. angustifolius* (19.5 ± 0.3%) > *Q. rubra* (19.1 ± 0.6%).

The seeds of oats (*A. strigosa*) began to germinate on the fifth day from the experiment onset. Seed germination capacity was high at 92–96%. The effect of leaf litter extracts on these *A. strigosa* seeds was weaker (in comparison with *V. cracca*): seed germination in these experiments was suppressed only by 1–2% as compared with the control.

#### 2.3.2. Assessment of the Allelopathic Activity of the Leaf Fall of Invasive Woody Species on the Growth of Test Plants

Extracts of all the studied invasive species demonstrated a significant (*p* < 0.01, Kruskal–Wallis test) negative influence on the hypocotyl length of vetch seedlings (Figure 4 and Figure 5). The appearance of the negative effect differed significantly between three “groups” of invasive species (*p* < 0.01 in each case, Mann–Whitney pairwise test). Thus, the leaf litter extracts of *Q. rubra* and *R. pseudoacacia* showed the strongest negative effect; the leaf litter extracts of *A. negundo* had an intermediate negative effect, and the negative effect of the leaf litter extracts of *A. altissima* and *E. angustifolia* was the weakest. Within these “groups”, the negative effects did not differ significantly (*p* > 0.05 in each case, Mann–Whitney pairwise test).

After 2 weeks, the oat (*A. strigosa*) seedlings had an average length of the most elongated root: 6.1 ± 0.5 cm in the control, and in the experimental groups, this parameter was as follows: leaf litter extract of *A. negundo*, 5.9 ± 0.4 cm; leaf litter extract of *A. altissima*, 5.4 ± 0.7 cm; leaf litter extract of *E. angustifolia*, 6.4 ± 0.5 cm; leaf litter extract of *Q. rubra*, 7.3 ± 0.8 cm; and leaf litter extract of *R. pseudoacacia*, 6.9 ± 0.4 cm (Figure 6). Therefore, the leaf litter extracts can act either as an oat (*A. strigosa*) root growth-inhibitor (groups 1–2) or as an oat (*A. strigosa*) root growth-stimulator (groups 3–5), depending on the species of a woody plant.

The leaf litter extracts of all the studied invasive woody species demonstrated a significant (*p* < 0.01, Kruskal–Wallis test) negative influence on the length of oat (*A. strigosa*) shoots (Figure 7). All tested extracts of the invasive species demonstrated significant effects in comparison with the control treatment (*p* < 0.01 in each case, Mann–Whitney pairwise test). The weakest negative effect was demonstrated by the leaf litter extract of *A. negundo*. The leaf litter extracts of the rest of the species (*A. altissima*, *E. angustifolia*, *Q. rubra*, and *R. pseudoacacia*) showed stronger negative effects, which did not differ significantly between most of them (*p* > 0.05 in each case, Mann–Whitney pairwise test), with the exception of a pair *A. altissima* and *R. pseudoacacia* (*p* < 0.01, Mann–Whitney pairwise test). In absolute values, the growth retardation of *A. strigosa* under the influence of the leaf litter of invasive species was less noticeable than the growth retardation of *V. cracca.*

A correlation analysis of the experiment results (Table 1) revealed that all groups of metabolites had a significantly negative relation with the growth of shoots of *A. strigosa*, but only the carbohydrate group demonstrated a significant relation for the growth of hypocotyls of *V. cracca*.

Thus, the leaf litter extracts of the most aggressive invasive woody plants have stronger negative effects on seed germination and seedling growth in dicots than in monocots. In the tested plant *V. cracca*, seed germination and growth inhibition by more than 50% were documented here, whereas in the tested plant *A. strigosa*, almost all seeds germinated, and the inhibition of growth did not exceed 31%, and in the roots, even a stimulatory effect of some assessed leaf litter extracts on root growth was noted. There was also a species-specific influence of leaf litter extracts on the tested crops. The leaf litter extracts of *Q. rubra* and *E. angustifolia* had the strongest inhibitory effect on vetch (*V. cracca*) seed germination capacity, whereas the leaf litter extracts of *A. negundo*, *R. pseudoacacia*, and *Q. rubra* had the greatest negative effect on vetch hypocotyls growth.

The leaf litter extracts of *A. negundo* and *A. altissima* inhibited the growth of oat (*A. strigosa*) roots, whereas the leaf litter extracts of *R. pseudoacacia*, *E. angustifolia*, and *Q. rubra* stimulated their growth. The negative influence of the leaf litter extract of *A. negundo* on the average length of oat (*A. strigosa*) of shoots can be regarded as weak, whereas the negative effects of experimental extracts from the other four tree species can be considered moderate (26–31%).

## 3. Discussion

The problems with vegetation recovery under invasive tree species can be explained by chemical legacy. Seed germination is the most important phase of the plant life cycle and is inhibited or delayed by allelochemicals [20]. In other studies, phytotoxic effects have been found to reduce the speed of seed germination [6,21,22]. It has been shown that the inhibition of seed germination is stronger in dicotyledonous plants than in cereals. These results are consistent with the absence of a grass layer under invasive tree species, in particular, under *A. negundo* and *Q. rubra*. The quality of leaf litter, namely, the unique compounds released from fallen leaves into the environment, which are characteristic of only certain plant species, affect microbial activity, the structure of the soil microbial community, and plant communities in the undercrown space of these species [23].

For instance, some authors have demonstrated that, at some concentrations, protocatechuic acid and ferulic acid can significantly reduce the parameters of early growth of seeds and seedlings [22,24]. The dominance of protocatechoic acid observed by us here in the leaf litter of *Q. rubra* confirms the results of the previous studies. The leaf litter extract of *Q. rubra* had the strongest allelopathic effect on the two tested crops (*V. cracca* and *A. strigosa*). In addition, the strong inhibitory properties of the leaf litter extract of *Q. rubra* may be due to its high concentration of condensed tannins because they contribute to the formation of stable protein–tannin complexes and affect carbon and nitrogen biogeochemical cycles, thereby reducing the availability of nitrogen [11].

Our results on macroelement concentrations in the leaf litter are somewhat consistent with those reported by Medina-Villar et al. [10], who showed that the leaf litter of the invasive woody plants *A. altissima* and *R. pseudoacacia* contains twice as much nitrogen, phosphorus, and organic matter as the litter of the native species *Populus alba* in tugai ecosystems of Central Spain.

Furthermore, some data suggest that there is resorption of C, N, P, K, and Mg during leaf senescence; this process is more active in leaves with lower nutrient concentrations [25]. On the other hand, it has been reported that the higher the concentration of Mg in the leaf litter of trees, the higher the leaching rate of the litter. For instance, this element is necessary for Mn-peroxidase activity, which is a lignin-degrading enzyme [26,27]. Overall, the same was observed in our study as well: we registered the lowest concentrations of C, P, K, Mg, and Mn in the slowest-decaying leaf litter (that of *Q. rubra*). Similar concentrations of Ca in the leaf litter among all the analyzed tree species suggest that different woody species can absorb this nutrient in similar amounts, and this element is retained in the leaves. Ca is an important structural element in plants, especially in cell walls, and tends to be resorbed to a lesser extent during aging than most other elements [20]. It has been noted in previous studies that high levels of Ca, Mg, K, C, and oxygen in the ash of leaf litter indicate the presence of oxalate and carbonate mineral deposits in them. A long time ago, E.I. Parfenova and E.A. Jarilova [28] found crystals of wavellite in the fallen leaves of oak; in this form, calcium oxalates are present in intracellular deposits. CaCO_3_ carbonates in the form of calcite and dolomite are present in slightly smaller amounts in extracellular and intercellular deposits. Earlier, these minerals were detected in the fallen leaves of *Moringa oleifera* [29], and in the ash of this species, a compound called fairchildite was identified, which can arise via thermal fusion of CaCO_3_ and K_2_CO_3_ [30].

It is often reported that laboratory bioassays involving filter paper overestimate the ability of allelochemicals to influence plant germination and growth parameters [20]. Indeed, the effects of less inhibitory allelochemicals disappear or even become positive in natural soil [20] because microbial communities strongly influence the persistence, availability, and biological activity of allelochemicals through the degradation and transformation of these compounds [31]. The findings of our paper indicate that leaf litter organic compounds are the most potent allelochemicals, which are mostly released during leaching and affect plants in the undercrown space of invasive trees. The leaf litter of invasive woody species is rich in allelochemical substances; consequently, its removal may facilitate the natural recovery of plant communities in the undercrown space. A thick layer of ground litter will negatively affect natural recovery and seedling numbers, although it is known that ground litter is nevertheless necessary because it plays a major role in the regulation of nutrient cycling in soil.

## 4. Materials and Methods

### 4.1. Plant Material

We used leaf litter of the following woody alien species: *A. altissima*, *Q. rubra*, *A. negundo*, *R. pseudoacacia,* and *E. angustifolia.* All these species are among the TOP 100 most aggressive invasive species in Europe [32]. The sampling site was located in the initial invasive populations on the abandoned land in the vicinity of the city of Nitra (central Slovakia; 48°18′ N, 18°05′ E). The sample of each invasive species was composed of field material from 5 points within the sampling site. Thereby, the joint sample for the particular invasive species included leaf litter material from different specimens of the chosen species. The plant material was collected in September 2020. Leaf litter of woody alien species fell to the ground and stayed on the soil for half a month. The samples of collected leaf litter were air-dried in the shade, ground with an electric grinder LZM-M1 (Laboratoroff, Voronezh, Russia), and passed through a 2 mm sieve.

### 4.2. The Profiling of Secondary Metabolites

One hundred milligrams of leaf litter powder for each sample were extracted with 1 mL deionized water using an ultrasonic bath (Saphur, Moscow, Russia) at room temperature for 30 min. Then, 750 μL chloroform and 800 μL deionized water were added to the solution and mixed well. Subsequently, the solution was centrifuged at 5000× *g* for 10 min Sigma 3-18KHS, (Sigma, Deisenhofen, Germany). A total of 200 mL of the centrifugate was evaporated to dryness under a helium stream. Derivation (trimethylsilylation) was performed with N,O-Bis (trimethylsilyl) trifluoroacetamide (BSTFA) following the method described by Marinova [33] and Han [34]. The silylation with BSTFA lasted for 30 min at 100 °C.

The samples (1 μL) were injected in the split mode (1:10) and identified with a JMS-Q1050GC (JEOL Ltd., Tokyo, Japan) quadrupole mass spectrometer equipped with an Agilent 7890B gas chromatograph and a 7693 autosampler (Agilent Technologies Inc., Tokyo, Japan). GC separation was achieved using a DB-5HT capillary column (30 m × 0.25 mm ID × 0.25 μm film, Agilent Technologies Inc., Tokyo, Japan), with helium as the carrier gas at a flow rate of 1 mL/min, 250 °C injector and transfer line temperature, and full scan with a scan range 50–550 amu. Compounds in leachates were identified by comparing their retention times to the National Institute of Standards and Technology (NIST) mass spectra library in the spectrometer database. The probability of substance identification was 75% to 98%. The relative abundance (%) of each compound was calculated by setting the total concentration of all components to 100% in an analyzed sample. The peak height of individual compounds was at least 0.1% of the indicator scale. All determinations were carried out in five-fold repetition.

### 4.3. The Profiling of Chemical Elements

The dried samples of leaf litter (2 g) were mineralized in a muffle furnace (Naberterm, Lilienthal, Germany) at 400 °C. The obtained ash was dispersed with ultrasonication at 18 kHz for 15 min. An even layer of the dispersed sample was applied to the analyzer’s stage covered with carbonic scotch.

The mineral (ash) composition was determined using an energy dispersive spectroscopy (EDS) analyzer combined with a JEOL JSM 6090 LA scanning electron microscope (Japan) in accordance with the methodology of Motyleva [35]. Spectra and element distribution data were obtained together with images using a raster electron microscope. The elemental composition was evaluated based on the weight percentage of 12 elements (O, C, P, S, K, Ca, Mg, Si, Al, Mn, Mo, and Zn) that were reliably identified. EDS was used for qualitative and quantitative analyses of elements in the samples with the help of the X-ray spectra acquired with electronic beam scanning of the observed image. Six measurements were taken for each ash sample. The analyzed area was 3 mm^2^, and the scanned area was at least 12 µm^2^. The average quadratic deviation did not exceed 1.2–6.9%.

### 4.4. An Assay of the Allelopathic Activity of Leaf Litter toward Two Herbaceous Plants

As test plants exposed to the leaf litter aqueous extracts, we used vetch seeds (*V. cracca*) and oat seeds (*A. strigosa*) from a lawn grass mixture that is used for the reclamation of disturbed urban and industrial lands.

To assess the test plants’ growth, vetch seeds (*V. cracca*) were grown in Petri dishes (25 seeds per dish) on moist filter paper for 7 days; the total amount of a metabolite solution (see below) per dish was 25 mL. Oat seeds (*A. strigosa*) (25 seeds per laboratory beaker) were grown on moist filter paper in laboratory beakers for 14 days, and the total amount of the metabolite solution per beaker was 50 mL.

The filter paper was moistened with water (control) or with the leaf litter extracts of the studied invasive species at 1:10 dilution in water. In experiments with *V. cracca*, the hypocotyl length of seedlings was measured. In experiments with *A. strigosa*, the shoot length and the length of the longest root were measured.

### 4.5. Statistical Analysis

Multivariate comparisons of metabolite and element profiles among invasive species were conducted using permutation MANOVA (non-parametric MANOVA). For visualization of metabolite and element profiles of the samples, we applied non-metrical multidimensional scaling (nMDS) on Euclidean distances between relative contents of substances (elements). The normality of chemical element content distribution in samples was tested using a Shapiro–Wilk test. Further testing of the significance of differences among species by the content of individual chemical elements was carried out using ANOVA in the case of data normality or using a Kruskal–Wallis test in the opposite case. Average values in the text are shown as the mean ± standard error.

In the germination experiments, the normality of the result value distributions was tested using a Shapiro–Wilk test. As normality was rejected (*p* < 0.05) for some replications and for some experiment treatments (invasive species), we applied the non-parametric Kruskal–Wallis test for multisample comparisons and the non-parametric Mann–Whitney pair-wise test. Since no significant differences were found among replicates within each experiment treatment, further between-treatment comparisons with the Kruskal–Wallis test were carried out for joint samples [36]. Due to the non-normality of the result distributions, we applied non-parametric Spearman’s correlation coefficient for correlation analysis.

All statistical analyses were conducted in R [37]. Multivariate analyses (ordination and PerMANOVA) were conducted with the R package “vegan” [38]. Visualizations were plotted with the R package “ggplot2” [39].

## 5. Conclusions

The most potent allelochemicals are the leaf litter organic compounds of alien woody species included in the TOP 100 most aggressive invasive species of Europe including *A. altissima*, *Q. rubra*, *A. negundo*, *R. pseudoacacia*, and *E. angustifolia*. The leaf litter aqueous extracts of all five invasive woody plants have a negative effect on seed germination and initial growth of *V. cracca* and *A. strigosa* used for the reclamation of disturbed urban and industrial lands.

Aqueous extracts of leaf litter have a stronger negative effect on *V. cracca* than on *A. strigose*. In the tested plant *V. cracca*, the death of seeds and inhibition of growth by more than 50% were noted, whereas in *A. strigosa*, almost all seeds germinated, but further growth was suppressed (by more than 30%). The allelopathic impact of the leaf litter extract of *Q. rubra* is determined mainly by the relative abundance of phenolic and aliphatic acids, the leaf litter extract of *R. pseudoacacia* is determined by carbohydrates and their derivatives and ester of fatty acid, and the leaf litter extract of *A. altissia* is determined by glycosides.

## Figures and Tables

**Figure 1 plants-12-04084-f001:**
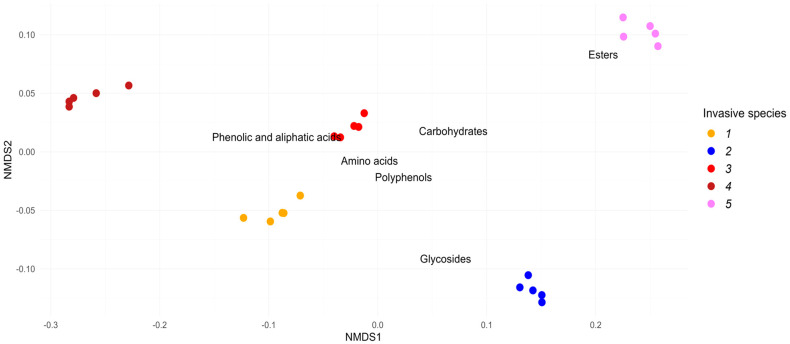
Ordination of species metabolite profiles of the leaf litter extracts of alien woody species (1—*Acer negundo*, 2—*Ailanthus altissima*, 3—*Elaeagnus angustifolia*, 4—*Quercus rubra*, 5—*Robinia pseudoacacia*) by 6 groups of metabolites (“Phenolic and aliphatic acids”, “Carbohydrates”, “Polyphenols”, “Esters”, “Glycosides”, and “Amino acids”) on 2 axes of reduced ordination space using nMDS. Color dots are samples (5 observations), and labels are descriptor scores (variables—metabolite groups). Both overall differences among species profiles and pairwise between-species differences were statistically significant (PerMANOVA *p* < 0.01 and *p* < 0.05 in each case, respectively).

**Figure 2 plants-12-04084-f002:**
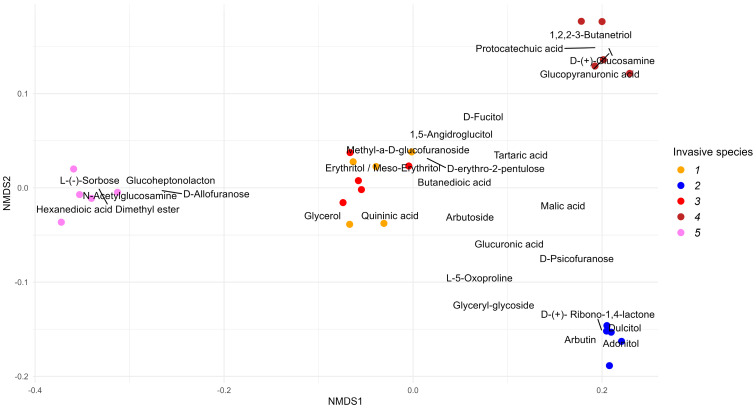
Ordination of species metabolite profiles of the leaf litter extract of alien woody species (1—*Acer negundo*, 2—*Ailanthus altissima*, 3—*Elaeagnus angustifolia*, 4—*Quercus rubra*, 5—*Robinia pseudoacacia*) by the most common and abundant individual metabolites on 2 axes of reduced ordination space using nMDS. Color dots are samples (5 observations), and labels are descriptor scores (variables—metabolites). The positions of some labels in the plot were randomly adjusted to avoid text clumping and to improve plot readability; the black connection lines point to the true label positions. Both overall differences among species profiles of the leaf litter extracts and pairwise between-species differences were statistically significant (PerMANOVA *p* < 0.01 and *p* < 0.05 in each case, respectively).

**Figure 3 plants-12-04084-f003:**
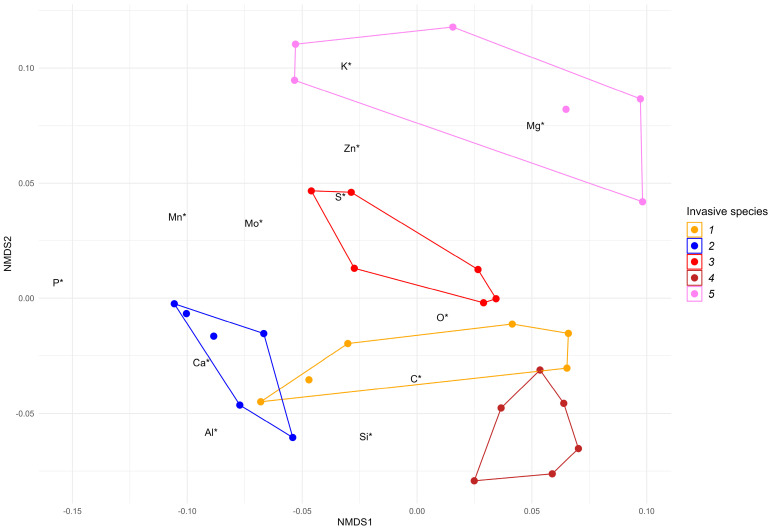
Ordination of chemical element profiles of the leaf litter of the studied invasive woody species ((1—*Acer negundo*, 2—*Ailanthus altissima*, 3—*Elaeagnus angustifolia*, 4—*Quercus rubra*, 5—*Robinia pseudoacacia*)) on 2 axes of reduced ordination space using nMDS. Color dots are samples (5 observations), and labels are descriptor scores (variables—chemical elements). * denotes chemical elements: C, O, K, Ca, Mg, Mn, Si, S, Al, Mo, Zn. The positions of labels in the plot were adjusted with a factor of 5 to avoid text clumping and to improve plot readability. Both overall differences among species profiles and pairwise between-species differences were statistically significant (PerMANOVA *p* < 0.01 and *p* < 0.05 in each case, respectively).

**Figure 4 plants-12-04084-f004:**
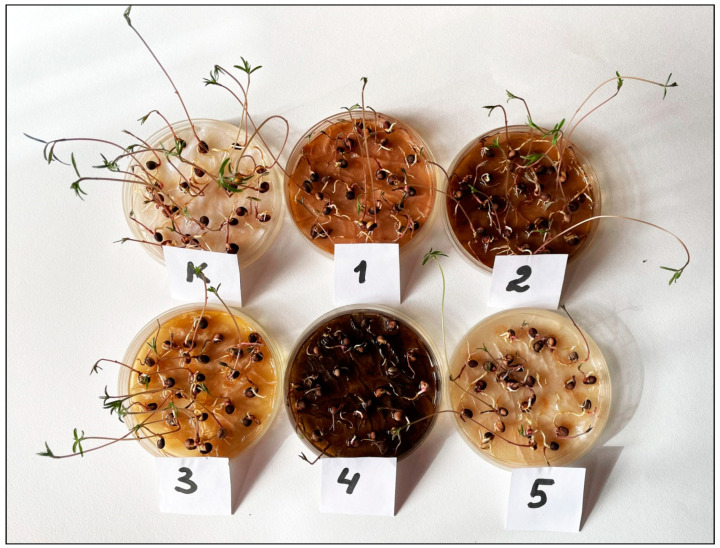
Effects of the leaf litter extracts of invasive woody species on the hypocotyl development of *Vicia cracca*. K—Control, 1—*Acer negundo*, 2—*Ailanthus altissima*, 3—*Elaeagnus angustifolia*, 4—*Quercus rubra*, and 5—*Robinia pseudoacacia*.

**Figure 5 plants-12-04084-f005:**
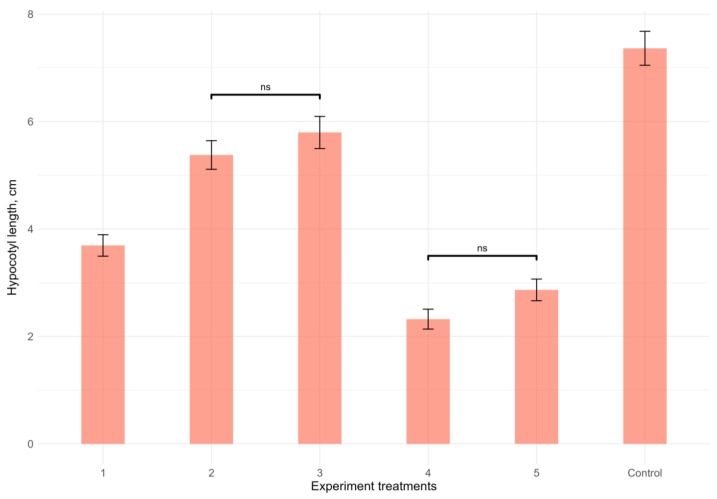
Effects of the leaf litter extracts of invasive woody species on the hypocotyl growth of *Vicia cracca* L. The overall difference was significant (*p* < 0.01, Kruskal–Wallis test). All pairwise differences were significant as well, except those designated in the plot as “ns” (not significant). Numbers of experimental treatments represent the leaf litter extracts of different species: 1—*Acer negundo*, 2—*Ailanthus altissima*, 3—*Elaeagnus angustifolia*, 4—*Quercus rubra*, and 5—*Robinia pseudoacacia*.

**Figure 6 plants-12-04084-f006:**
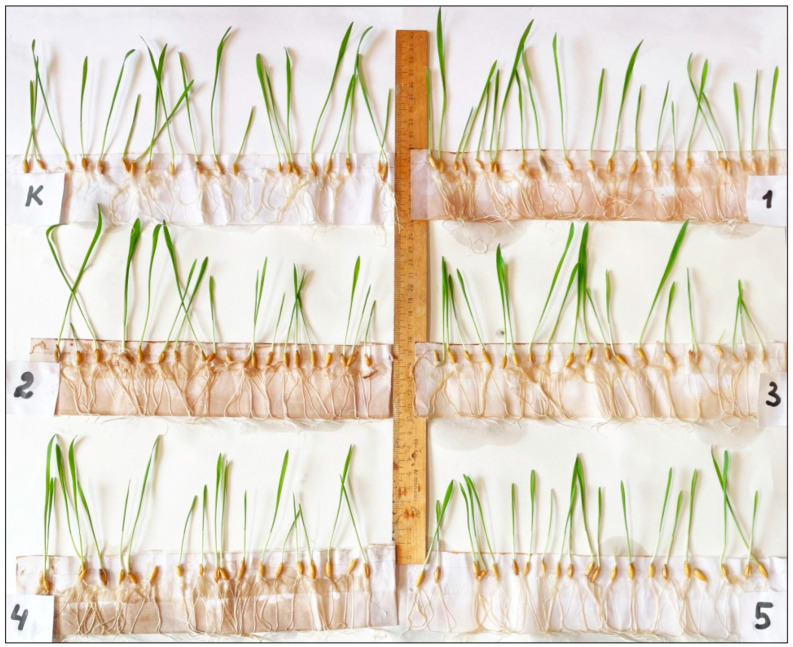
Effects of the leaf litter extracts of invasive woody species on the shoot development of *Avena strigosa*. K—control, 1—*Acer negundo*, 2—*Ailanthus altissima*, 3—*Elaeagnus angustifolia*, 4—*Quercus rubra*, and 5—*Robinia pseudoacacia*.

**Figure 7 plants-12-04084-f007:**
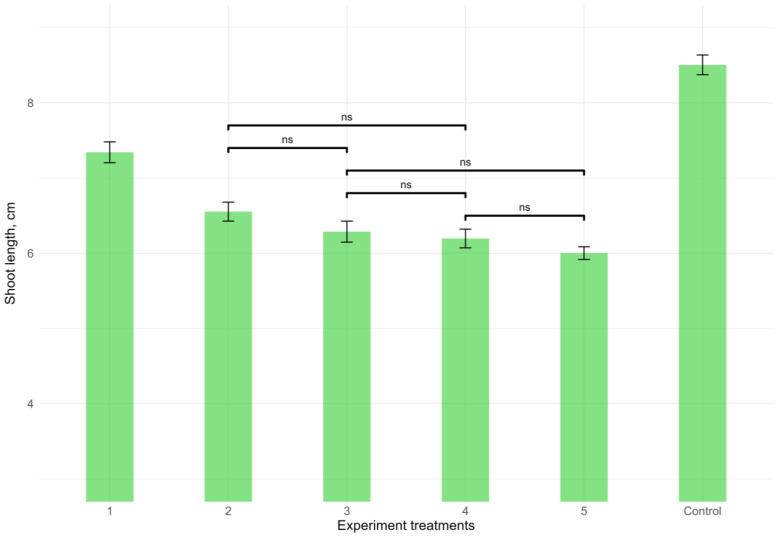
Effects of the leaf litter extracts of invasive woody species on the growth of shoots of *Avena strigosa*. The overall difference was significant (*p* < 0.01, Kruskal–Wallis test). All pairwise differences were significant as well, except those designated in the plot as “ns” (not significant). Numbers of experimental treatments represent the leaf litter extracts of different species: 1—*Acer negundo*, 2—*Ailanthus altissima*, 3—*Elaeagnus angustifolia*, 4—*Quercus rubra*, and 5—*Robinia pseudoacacia*.

**Table 1 plants-12-04084-t001:** Correlation analysis of growth experiment results for *A. strigosa* and *V. cracca*. Entries are Spearman’s correlation coefficients (lower triangle) and its *p*-values (upper triangle). Significant correlations between seedling length and metabolites are in bold.

		*p*-Values							
		Length (Vetch Hypocotyls)	Length (Oat Shoots)	Phenolic and Aliphatic Acids	Esters	Carbohydrates	Polyphenols	Glycosides	Amino Acids
Spearman’s Correlation Coefficients	Length (vetch hypocotyls)			<0.01	0.72	<0.01	0.82	0.95	0.41
	Length (oat shoot)			<0.01	<0.01	<0.01	<0.01	<0.01	<0.01
	Phenolic and aliphatic acids	**−0.41**	**−0.23**		<0.01	0.43	<0.01	<0.01	<0.01
	Esters	0.01	**−0.27**	−0.23		<0.01	<0.01	<0.01	<0.01
	Carbohydrates	**−0.28**	**−0.40**	0.03	0.64		<0.01	<0.01	<0.01
	Polyphenols	0.01	**−0.21**	0.43	0.32	0.26		<0.01	<0.01
	Glycosides	0.00	**−0.17**	0.14	0.52	0.49	0.71		<0.01
	Amino acids	−0.03	**−0.20**	0.31	0.7	0.14	0.54	0.49	

## Data Availability

Data is contained within the article and Appendix A.

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
