# Peer review of "Leaf Extracts of Invasive Woody Species Demonstrate Allelopathic Effects on the Growth of a Lawn Grass Mixture"

_plants, 2023, doi:10.3390/plants12244084_

Round 1

Reviewer 1 Report

Comments and Suggestions for Authors

The manuscript is interesting and well presented. However some minor improvement are needed.

Please, remove species authorities in the abstract and MM chapter. Apply the authorities and the species full names where they appear firstly in the text. Size letter vary in the text, please check up. Also, apply Latin names at least in brackets there where You firstly apply trivial English species names (e.g. where test herbaceous species are mentioned). Also You put firstly for example A. strigosa, and than in last chapter Avena strigosa. It should be opposite.

Please, also explain why did You chose the tested species (Viccia and Avena), and why litter from the autumn and not the fresh leaves? Why not the spring litter? 

Figures needs improvements. Both visual and explanation. Explanations needs to be so that it is clear to one that has not read full text. E.g. the bars explanation can not be tree species name but the type of extract, maybe betters with numbers; also clearly explain under the figures.

The same is also used in the text of the manuscript which is incorrect. You cannot say R. pseudoacacia affect but the extract of R. pseudoaccacia... etc.

Please pay attenetion to be precise and not colloquial.

The English looks like it is written by various persons, and partly it is hard to follow. Therefore it needs improvement.

Comments on the Quality of English Language

I think the MS is worthy of publication but I also think it can be improved.

Author Response

Dear reviewer, thank you very much for your careful reading of our article!

In the new version of our article we have corrected the Latin names of plants, edited the text, corrected the figures, explained why we use V. cracca and A. strigosa as test plants, investigated the autumn leaf litte .

Reviewer 2 Report

Comments and Suggestions for Authors

I suggest some minor corrections as follows:

Use the term extract instead of leaf litter where solutions were tested. At present, there are some mixed uses of the terms.

Use numbers, not words, when the number of compounds is described

Add standard deviation or standard error, and number of analyzed data (numerus) when describing the results

Replace seedling length in the figures with shoot (or hypocotyl in vicia)

To improve discussion, I suggest to differentiate the effects on seed germination and seedling growth.

Since you obtained so many results on chemical composition, the  discussion could be improved regarding different inhibitory effects of Different "groups" of compounds. Add possible mechanisms of their function when applicable.

Correct format of some references (i.e. some journal ar fully named some abbreviated...)

The corrections are marked in the revised paper.

Author Response

Dear reviewer, thank you very much for your careful reading of our article!

In the new version of our article we have made corrections - we use "extracts of leaf litter", added standard deviation to the data, use numbers instead of words, divided chapter 2.3 into two parts, corrected the format of references.